# Menopause education of healthcare professionals: A scoping review protocol

**C. Keye** [ID][1]*, **Murphy M**[2], **Saab MM**[2], **O' Driscoll M**[1]

1 Pharmaceutical Care Research Group, School of Pharmacy, University College Cork, Cork, Ireland,
2 Catherine McAuley School of Nursing and Midwifery, University College Cork, Cork, Ireland

* 125106949@umail.ucc.ie

## Abstract

The reported lack of adequate menopause care is a significant issue, leaving many women feeling isolated and unsupported as they navigate this transition. The increasing mainstream attention on menopause healthcare highlights the need to evaluate whether healthcare professionals are adequately educated and competent to provide effective care for menopausal women. The aim of this scoping review is to ascertain to what extent undergraduate and post registration healthcare professionals are educated regarding menopause throughout their training and subsequent careers, with a view to informing future educational initiatives. This scoping review will be carried out using the JBI Manual for Evidence Synthesis and reported using the Preferred Reporting Items for Systematic reviews and Meta-Analyses extension for scoping reviews checklist. It will include all healthcare professionals relevant to the provision of menopause care, including medical doctors, nurses, midwives, pharmacists, physiotherapists, and psychologists. The search will use the following databases: CINAHL; MEDLINE; EMBASE; APA PsycINFO; ERIC; and Academic Search Complete. Grey literature will be searched using Google Scholar and Open Grey. An academic librarian has been contacted to assist in the development of the search strategy. Covidence online software will be used to manage the screening and data extraction process. Purposely developed data extraction tables will be created by the reviewers. The data will be synthesised and reported narratively.

## Introduction

Menopause is defined as the last menstrual period, with perimenopause being the time leading up to the final menses and post-menopause being one year since the final period [1–3]. The average age of menopause is 51 years [4]. Symptoms of menopause can begin 7–10 years before cessation of menses and can last for up to 12 years post menopause, meaning women can spend up to 22 years experiencing the impacts of menopause [5].

**Data availability statement:** No datasets were generated or analysed during the current study. All relevant data from this study will be made available upon study completion.

**Funding:** The authors would like to acknowledge that C. Keye is in receipt of a PhD scholarship from the School of Pharmacy, University College Cork, Cork, Ireland.

**Competing interests:** The authors have declared that no competing interests exist.

Research shows many women feel unprepared, unsupported, and receive poor practitioner care during menopause transition [6–8]. It is recommended that an understanding of menopause physiology, signs and symptoms, and impact of menopause are essential for creating an individualised patient management plan [9]. However, clinicians have reported a lack of understanding and treatment of menopause [9], with training gaps leading to half of healthcare professionals (HCPs) studied not being aware of contraindications to prescribing menopausal hormonal treatment [10,11]. Despite the fact that there is no limitation to the length of time women should remain on hormone replacement therapy (HRT) and that the British Menopause Society recommend persistent symptoms usually overrule the risks [12], 2023 data from the United Kingdom (UK) shows that only approximately 15% of women aged between 45–64 use HRT [13]. This can be attributed to several factors, including a shortage of HRT [14], lower prescribing rate in deprived areas [15] and probably most importantly, the impact of the 2002 Women's Health Initiative trial [16]. This large, long-term study had a profoundly negative impact on the perception and prescribing of HRT. Initial findings in that study suggested that HRT posed more risks than benefits, highlighting increased risks of breast cancer and cardiovascular disease, leading to a significant decline in HRT use, as both patients and healthcare providers became wary of its potential harms [16]. However, it was later shown that this study's analysis was flawed. It has since been reanalysed, demonstrating that HRT, when initiated at the right time and for the right population, can have significant benefits, including symptom relief, cardiovascular protection, and improved bone health [16].

In 2014, the European Menopause and Andropause Society provided guidance on the importance of medical undergraduate training in menopause [17]. This statement was updated in 2022 to include all HCPs at all stages of their careers, ensuring that knowledge about reproductive ageing and menopause is integrated into all healthcare education and training programs [17]. The British Menopause Society vision statement also agrees that menopause care is multi-disciplinary, and that it is necessary for all HCPs, both in primary and secondary care, to have access to sufficient menopause education [18]

To the best of our knowledge, and based on a preliminary search of CINAHL, Medline, and Embase, only one existing scoping review was found, which explored the menopause education of. HCPs [19]. However, this scoping review only included doctors and did not include other allied HCPs. Therefore, the overarching aim of the current scoping review is to assess the extent of the literature on HCP education on menopause care.

The review questions were developed using the population, concept, and context (PCC) framework as recommended by the JBI Manual for Evidence Synthesis [16] as follows:

1. What menopause educational programmes are available for HCPs?

2. Which HCP(s) have received menopause education?

3. How has menopause education been delivered?

4. Where and in what settings has menopause education been delivered?

## Methods

This scoping review will be conducted in accordance with the JBI methodology for scoping reviews [20], and will be reported using the Preferred Reporting Items for Systematic reviews and Meta-Analyses extension for scoping reviews (PRISMA-ScR), ensuring the scoping review is carried out rigorously, and is transparent and reproducible, enhancing the credibility and utility of the findings (Appendix S1) [21].

### 1. Inclusion criteria

Review inclusion criteria have been informed by the PCC framework as follows:

#### Population

- All HCPs who provide menopause care to women, inclusive of nurses, midwives, doctors, physiotherapists, pharmacists, and psychologists.

- Undergraduate, postgraduate students and qualified HCPs undertaking Continuing Professional Development.

#### Concept

This review is focused on exploring the breadth of evidence available on the types of HCPs' menopause education and training delivered. This review will help identify key concepts, definitions, important learning outcomes, as well as identify specific indicators that can be used to shape the development of research design and a future comprehensive educational model.

#### Context

The review will include all HCPs who are involved in the provision of menopause-related care in both primary and secondary healthcare settings. There will be no geographical, language, or date limitations placed on the review. All primary research papers relating to menopause education provision will be included. This scoping review will consider quantitative studies such as experimental or quasi-experimental study designs, including randomised and non-randomised controlled trials. This review will also include descriptive observational studies. Qualitative studies will be included with a focus on, but not limited to, study designs such as phenomenology, grounded theory, and qualitative description. Articles, reviews, abstracts, and opinion pieces will not be considered for inclusion.

### 2. Search strategy

A preliminary search was conducted in MEDLINE to identify relevant free-text and subject heading terms. Keywords were determined through discussions with an academic librarian and the co-authors, and further terms were identified by examining literature during the scoping phase of the search. Using these terms, a comprehensive search strategy for MEDLINE was developed (S2 Appendix). The search strategy will be developed further by all the authors in conjunction with the academic librarian and will aim to locate all relevant published and unpublished studies across CINAHL, MEDLINE, EMBASE, APA PsycINFO, ERIC, and Academic Search Complete. The final search strategy will be included in the scoping review. Of note, journals by professional organisations, such as the International Federation of Gynaecology and Obstetrics, British Menopause Society, International Menopause Society and European Menopause and Andropause Society, are indexed in at least one of the included electronic databases. In addition, we will search the following websites for unpublicised sources: EQUATOR Network (https://www.equator-network.org/); Centre for Evidence-Based Medicine (https://www.cebm.net/); Centre for Reviews and Dissemination (https://www.york.ac.uk/crd/); Critical Appraisal Skills Programme (https://casp-uk.net/); McMaster's Centre for Evidence-Based Medicine (https://hslmcmaster.libguides.com/c.php?g=306765&p=2044668); Evidence Synthesis International (https://evidencesynthesis.org/); JBI (https://JBI.global/);

and Campbell Collaboration (Campbell Collaboration (https://campbellcollaboration.org/). We will also conduct a hand search of the included articles to identify any relevant studies [22].

### 3. Study selection

Firstly, titles and abstracts will be screened against the predefined inclusion criteria. Documents that meet the criteria or require further clarification will advance to the full-text screening stage. Secondly, full-text screening will be conducted to thoroughly assess the full details of the remaining documents. Those that meet the criteria will be included in the review, while the remaining will be excluded, with the reason for exclusion documented. Thirdly, the reference list of all included papers will be hand-searched to source any additional relevant papers for inclusion. A PRISMA [23] flowchart will be used to illustrate the study screening process. Covidence online software will be used to assist in the automatic deletion of duplicated articles, with an additional manual screening for these conducted [24]. Title, abstract, and full text screenings will be conducted by two reviewers independently. Any disagreements that arise between the reviewers at each stage of the selection process will be resolved by a third independent reviewer.

### 4. Data extraction

Two data extraction tables have been drafted to collect all data of relevance to the review (S3 Appendix, S4 Appendix). The use of the data extraction tools reduces transcription errors, ensures all relevant information is collected, assists in checking accuracy and acts as a record of information collected [25]. Data extraction tables will be checked and validated by a second reviewer in line with best practice to reduce extraction errors [26]. The reviewers will firstly record general information such as the author(s), publication year, the country and setting of the research, as well as the population size, participants' profession, aims of the study and methodology (S3 Appendix). A second data extraction table has been developed to extrapolate specific details around the education provided to address the aims of the review (S4 Appendix). The draft data extraction tables will be modified and revised as necessary during the process of extracting data from each included evidence source. Modifications will be detailed in the scoping review.

### 5. Data analysis and presentation

Data will be synthesised by addressing the four key questions outlined in the review. By organising the findings into these categories, we will visualise relationships between the concepts, highlighting areas where consensus/contradictions exist to identify any gaps in the literature. The results will be reported narratively with diagrams, charts, or narrative maps to show relationships between themes. We will include descriptive statistics as appropriate. As this scoping review progresses, emergent themes or concepts not initially anticipated may arise. Given the iterative nature of scoping review methodologies [20,21], the framework will remain flexible to accommodate new insights. Any modifications to the structure or approach will be documented in the final review, ensuring transparency.

### Limitations

The authors will do their utmost to reduce study selection bias by using translation software to include non-English papers and optimising the search strategy to identify all relevant papers. There may also be limitations regarding searching grey literature, with some papers not being identified in the search due to a lack of controlled vocabulary.

### Conclusion

This scoping review aims to identify gaps in the literature to provide a platform for further evidence-based research on HCPs menopause education. This will subsequently serve to inform the development of a framework for future studies on service planning and delivery of menopause education for HCPs'.This article describes a protocol for a scoping review. The full results and any associated datasets will be made available in a subsequent publication upon completion of the scoping review.

## Supporting information

**S1 File. PRISMA protocol checklist.**
(DOCX)

**S2Appendix. Search strategy.**
(DOCX)

**S3 Appendix. Data extraction table 1.**
(DOCX)

**S4 Appendix. Data extractions table 2.**
(DOCX)

## Acknowledgments

The authors would like to acknowledge Siobhan Bowman, Academic Librarian, University College Cork, for assisting in the development of the search strategy for this scoping review.

## Author contributions

**Conceptualization:** Catriona Keye, Saab MM.

**Formal analysis:** Catriona Keye, Murphy M, O' Driscoll M.

**Methodology:** Catriona Keye, Saab MM, Murphy M, O' Driscoll M.

**Supervision:** Saab MM, Murphy M, O' Driscoll M.

**Visualization:** Catriona Keye, Saab MM, Murphy M.

**Writing – original draft:** Catriona Keye.

**Writing – review & editing:** Saab MM, Murphy M, O' Driscoll M.

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
