## [Decision Letter · Decision Letter 0]

15 Aug 2025

Dear Dr. Keye

Thank you for submitting your manuscript to PLOS ONE. After careful consideration, we feel that it has merit but does not fully meet PLOS ONE’s publication criteria as it currently stands. Therefore, we invite you to submit a revised version of the manuscript that addresses the points raised during the review process.

We look forward to receiving your revised manuscript.

Kind regards,

Fereshteh Behmanesh, PhD

Academic Editor

PLOS ONE

**Journal Requirements:**

1. When submitting your revision, we need you to address these additional requirements. Please ensure that your manuscript meets PLOS ONE's style requirements, including those for file naming. The PLOS ONE style templates can be found at https://journals.plos.org/plosone/s/file?id=wjVg/PLOSOne_formatting_sample_main_body.pdf and https://journals.plos.org/plosone/s/file?id=ba62/PLOSOne_formatting_sample_title_authors_affiliations.pdf 2. Thank you for stating the following financial disclosure: The authors would like to acknowledge that C. Keye is in receipt of a PhD scholarship from the School of Pharmacy, University College Cork, Cork, Ireland.    Please state what role the funders took in the study.  If the funders had no role, please state: "The funders had no role in study design, data collection and analysis, decision to publish, or preparation of the manuscript." If this statement is not correct you must amend it as needed. Please include this amended Role of Funder statement in your cover letter; we will change the online submission form on your behalf. 3. When completing the data availability statement of the submission form, you indicated that you will make your data available on acceptance. We strongly recommend all authors decide on a data sharing plan before acceptance, as the process can be lengthy and hold up publication timelines. Please note that, though access restrictions are acceptable now, your entire data will need to be made freely accessible if your manuscript is accepted for publication. This policy applies to all data except where public deposition would breach compliance with the protocol approved by your research ethics board. If you are unable to adhere to our open data policy, please kindly revise your statement to explain your reasoning and we will seek the editor's input on an exemption. Please be assured that, once you have provided your new statement, the assessment of your exemption will not hold up the peer review process. 4. If the reviewer comments include a recommendation to cite specific previously published works, please review and evaluate these publications to determine whether they are relevant and should be cited. There is no requirement to cite these works unless the editor has indicated otherwise. 

**Additional Editor Comments:**

**Reviewer 1**

In the search strategy, it is advisable to also search citing articles of known studies, so I would include this in the strategy. Also, with Open Grey not having been updated recently (archived in 2021), instead of searching this repository, I would suggest instead searching the websites of known organizations invovled in menopause and/or medical and health care professional education. For example, FIGO and its members or the Menopause Society. You can likely use the peer-reviewed literature to find references to useful groups.

Will the searches be included in the appendix of the final paper or archived somewhere else? This should be clear.

**Reviewer 2**

In the search phase, consider including the following websites to help identify pertinent unpublished sources.

EQUATOR Network (https://www.equator-network.org/);

Centre for Evidence-Based Medicine (https://www.cebm.net/);

Centre for Reviews and Dissemination (https://www.york.ac.uk/crd/);

Critical Appraisal Skills Programme (https://casp-uk.net/);

McMaster's Centre for Evidence-Based Medicine (https://hslmcmaster.libguides.com/c.php?g=306765&p=2044668);

Evidence Synthesis International (https://evidencesynthesis.org/);

JBI (https://JBI.global/);

Campbell Collaboration (https://campbellcollaboration.org/).

Reviewers' comments:

**Comments to the Author**

1. Does the manuscript provide a valid rationale for the proposed study, with clearly identified and justified research questions?

Reviewer #1: Yes

Reviewer #2: Yes

2. Is the protocol technically sound and planned in a manner that will lead to a meaningful outcome and allow testing the stated hypotheses?

Reviewer #1: Yes

Reviewer #2: Yes

3. Is the methodology feasible and described in sufficient detail to allow the work to be replicable?

Reviewer #1: Yes

Reviewer #2: Yes

4. Have the authors described where all data underlying the findings will be made available when the study is complete?

Reviewer #1: No

Reviewer #2: Yes

5. Is the manuscript presented in an intelligible fashion and written in standard English?

Reviewer #1: Yes

Reviewer #2: Yes

You may also provide optional suggestions and comments to authors that they might find helpful in planning their study.

**Reviewer #1:**  Thank you for the opportunity to review this protocol. It is well written according to the JBI standard and needs very few minor revisions.

In the search strategy, it is advisable to also search citing articles of known studies, so I would include this in the strategy. Also, with Open Grey not having been updated recently (archived in 2021), instead of searching this repository, I would suggest instead searching the websites of known organizations invovled in menopause and/or medical and health care professional education. For example, FIGO and its members or the Menopause Society. You can likely use the peer-reviewed literature to find references to useful groups.

Will the searches be included in the appendix of the final paper or archived somewhere else? This should be clear.

**Reviewer #2: ** In the search phase, consider including the following websites to help identify pertinent unpublished sources.

EQUATOR Network (https://www.equator-network.org/);

Centre for Evidence-Based Medicine (https://www.cebm.net/);

Centre for Reviews and Dissemination (https://www.york.ac.uk/crd/);

Critical Appraisal Skills Programme (https://casp-uk.net/);

McMaster's Centre for Evidence-Based Medicine (https://hslmcmaster.libguides.com/c.php?g=306765&p=2044668);

Evidence Synthesis International (https://evidencesynthesis.org/);

JBI (https://JBI.global/);

Campbell Collaboration (https://campbellcollaboration.org/).

**Do you want your identity to be public for this peer review?** For information about this choice, including consent withdrawal, please see our Privacy Policy

Reviewer #1: **Yes: ** Amanda Ross-White

Reviewer #2: No

---

## [Author Response · Author response to Decision Letter 1]

27 Aug 2025

Hi, Thank you for your feedback. We have attached a document in response you your recommendations.

---

## [Decision Letter · Decision Letter 1]

20 Oct 2025

Menopause education of healthcare professionals: A scoping review protocol

PONE-D-25-24077R1

Dear Dr. Catriona Keye,

We’re pleased to inform you that your manuscript has been judged scientifically suitable for publication and will be formally accepted for publication once it meets all outstanding technical requirements.

Kind regards,

Fereshteh Behmanesh, PhD

Academic Editor

PLOS ONE

Additional Editor Comments (optional):

Reviewers' comments:

Reviewer's Responses to Questions

**Comments to the Author**

1. Does the manuscript provide a valid rationale for the proposed study, with clearly identified and justified research questions?

Reviewer #2: Yes

2. Is the protocol technically sound and planned in a manner that will lead to a meaningful outcome and allow testing the stated hypotheses?

Reviewer #2: Yes

3. Is the methodology feasible and described in sufficient detail to allow the work to be replicable?

Reviewer #2: Yes

4. Have the authors described where all data underlying the findings will be made available when the study is complete?

Reviewer #2: Yes

5. Is the manuscript presented in an intelligible fashion and written in standard English?

Reviewer #2: Yes

You may also provide optional suggestions and comments to authors that they might find helpful in planning their study.

Reviewer #2: I have no further comments, as all previous feedback has been thoughtfully incorporated. Congratulations!

**Do you want your identity to be public for this peer review?** For information about this choice, including consent withdrawal, please see our Privacy Policy

Reviewer #2: No

---

## [Editor Report · Acceptance letter]

PONE-D-25-24077R1

PLOS ONE

Dear Dr. Keye,

I'm pleased to inform you that your manuscript has been deemed suitable for publication in PLOS ONE. Congratulations! Your manuscript is now being handed over to our production team.

Kind regards,

on behalf of

Dr. Fereshteh Behmanesh

Academic Editor

PLOS ONE